**Perspective**

# Move the night way: how can physical activity facilitate adaptation to shift work?
Dayna F. Easton [1] ✉, Charlotte C. Gupta[1], Grace E. Vincent[1] & Sally A. Ferguson[2]

Shift work, involving night work, leads to impaired sleep, cognition, health and wellbeing, and an increased risk of occupational incidents. Current countermeasures include circadian adaptation to phase shift circadian biomarkers. However, evidence of real-world circadian adaptation is found primarily in occupations where light exposure is readily controlled. Despite this, non-photic adaptation to shift work remains under researched. Other markers of shift work adaptation exist (e.g., improvements in cognition and wellbeing outcomes) but are relatively unexplored. Timeframes for shift work adaptation involve changes which occur over a block of shifts, or over a shift working career. We propose an additional shift work adaptation timeframe exists which encompasses acute within shift changes in markers of adaptation. We also propose that physical activity might be an accessible and cost-effective countermeasure that could influence multiple markers of adaptation across three timeframes (Within Shift, Within Block, Within Work-span). Finally, practical considerations for shift workers, shift work industries and future research are identified.

The growing demands for 24 h services mean that non-standard work hours are the norm for a significant proportion of the Western workforce[1,2]. Approximately 14.8% of Australian employees reported undertaking shift work[3], while America and the United Kingdom report 17% and 14%, respectively[4,5]. Work shifts that occur outside the standard 9am–5pm workday oppose intrinsic physiological sleep-wake cues and require workers to maintain alertness when sleep drive is high[6]. Working during the biological night is associated with many negative repercussions for health, sleep and cognitive functioning[7–9]. Such negative consequences of shift work result from circadian misalignment, which broadly is a term that refers to the mismatch between the timing of sleep and wake, the light dark cycle, and other circadian rhythms (i.e., behavioural, physiological, hormonal, cellular)[10]. As such, misalignment can be internal (i.e., rhythms not aligned with each other) and/or external (i.e., rhythms not aligned with environment) which has implications for health and wellbeing.

Circadian misalignment be identified through circulating circadian hormones, like the relative melatonin and cortisol levels, and also observed at the molecular level through RNA and mRNA expression[10–12]. For instance, one study observed significant differences in circadian rhythms of gene expression patterns between night shift and day shift nurses[10]. The molecular and cellular impacts of circadian misalignment are thought to contribute to disease pathogenesis in shift workers[10,12]. The chronic inflammation of immune cells[12], and the disruption of the regulatory role sleep plays in immune function, also impair immunity[13] and increase vulnerability to infection[14]. The molecular changes arising from circadian misalignment due to shift work therefore increase risk for adverse health outcomes.

Circadian misalignment and partial chronic sleep deprivation also impacts many aspects of cognitive functioning, such as attention, reaction time, and visual processing speed, which are significantly impaired in regular night shift workers[15,16]. Deficits in cognitive functioning increase the chances of accidents and injuries while on shift[17,18] and are particularly dangerous for those in safety-sensitive and high-risk occupations. A review of shift work in safety-sensitive occupations found that lapses in attention and impaired decision-making significantly increased the occurrence of accidents during the night shift, such as needle-related injuries for healthcare workers[19] and preventable vehicle crashes for police officers[6,20]. Mitigating cognitive and safety risks is essential as night work has benefits for individuals and industries. Long blocks of night work are economically advantageous, particularly for operations such as mining or solar plant stations, where frequently flying staff in and out to maintain plant operations is not feasible[21]. On the other hand, employees often view greater work availability, salary, and family event attendance as an important trade-off for reduced sleep hours and poor sleep quality during their night shift rotations[22].

The total economic burden of shift work-related injuries due to reduced attentional capacity is difficult to determine. However, a report highlights that inadequate sleep in Australian workers in 2016-17 imposed

[1]Appleton Institute, School of Health, Medical and Applied Sciences, Central Queensland University, Wayville, SA, Australia. [2]Appleton Institute, Central Queensland University, Wayville, SA, Australia. ✉e-mail: d.f.easton@cqu.edu.au

financial losses of $26.2 billion and well-being losses (defined as loss of quality of life and leisure, and burden of disease) of $40.1 billion[23]. Given that shift work populations obtain inadequate sleep compared to other occupational groups[24], shift workers likely contribute significantly to these numbers[23]. A study of Australian on-call workers sheds further light on the economic burden for the shift working population given the similarities in non-traditional working hours in on-call work and shift work[25]. The authors highlight that the cost of inadequate sleep and workplace injury in on-call workers was estimated to be $2.25 billion annually, and related primarily to workplace compensation, reduced productivity and staff turnover[25]. Thus, the cost of inadequate sleep, workplace injury and well-being in occupational groups engaged in shift work or other non-standard hours is significant.

In addition to improving occupational safety, improving employee well-being also has important downstream impacts. Better well-being and mood are associated with improved job-satisfaction[26], physical and mental health[27], immune function[28] and lower stress reactivity[29]. Shift work has direct and indirect effects on biological, psychological and psychosocial systems[27]. For instance, the negative mood impacts of night shift heighten family conflict and stress, and some nurses consider this stress to be a strong factor impacting physical health[30]. However, better familial relationships are associated with a greater sense of resilience, self-esteem and life satisfaction[31]. Thus, night shift directly impacts psychological well-being and indirectly impacts familial relationships, resilience and overall coping with shift work[30]. Further, the relationship between job stress, interpersonal conflict and reduced psychological well-being contributes to staff turnover and absenteeism within the health sector[32]. Coping through absenteeism is common and is exacerbated by stressful working environments and workplace dissatisfaction, particularly among shift-working nurses[33]. However, this coping strategy has significant economic impacts, with the impact of absenteeism, presenteeism (attending work while ill or disengaged, resulting in reduced productivity) and labour turnover in the United Kingdom estimated to be £53-56 billion in 2020-21[34]. The report indicated that the public health sector made up 11% of survey respondents, second to that of education (14%) and other private sector services (14%)[34]. Thus, the intersection between biopsychosocial factors results in a feedback loop between employee, employer and economic burden, with shift workers and shift work industries significantly impacted.

To mitigate the negative effects of shift work on health, occupational risk, sleep and well-being, tailored strategies are needed. To date, these strategies have largely involved a focus on circadian adaptation. Circadian adaptation is typically defined as the process by which rhythms move to partially or fully match a new externally imposed schedule (e.g., time zone when travelling, or night shift when working)—otherwise referred to as phase shifting. Circadian phase shifts are defined and measured using markers of timing (e.g., onset, offset, peak, nadir) of core physiological rhythms, such as sleep/wake, melatonin, core body temperature and cortisol[35–38]. Importantly, evidence of circadian adaptation in naturalistic environments is largely found in occupations where ambient light exposure and avoidance are readily controlled, such as offshore oil installations[39,40]. In other environments, like hospital wards[41] or onshore mining sites[42], circadian adaptation is challenging. An alternative strategy is to use non-photic interventions to support adaptation, and to use markers of adaptation that might be beneficial for health and safety in shift workers other than circadian rhythms. For instance, adaptation to shift work has been reported in terms of improved physical[43] and mental health outcomes[27], or through reduced interpersonal conflicts[30]. Thus, we propose that shift work adaptation be viewed through a holistic lens that includes all biopsychosocial markers of adaptation and is defined as positive changes in markers that help shift workers to cope with irregular work hours (e.g., improved mood, sleep, performance, and job satisfaction).

Shift work adaptation has been referred to as adjustment[44–46] or tolerance[47–49] and can occur over both a "short-term" and "long-term" period[44–47]. For instance, Folkard et al.[46] defined short-term changes in circadian rhythms as occurring "over a period of successive (night) shifts" while long-term changes were defined as occurring "only after considerable experience of shift work"[46]. We posit that an additional timeframe of shift work adaptation exists which includes acute "within shift" changes in markers of adaptation such as performance and alertness. Thus, we aim to discuss several markers of shift work adaptation, including circadian adaptation, and three timeframes in which shift work adaptation occurs (See Fig. 1). Further, we propose physical activity as a potential strategy which may support markers of shift work adaptation across all three timeframes (Within Shift, Within Block, and Within Work-span).

## Circadian Adaptation to Shift Work

The circadian system is comprised of a central clock located in the suprachiasmatic nucleus of the hypothalamus, connected to other endogenous peripheral oscillators[50,51]. The cellular and molecular interactions of these central and peripheral oscillators collectively generate circadian rhythms and synchronise rhythms to the light-dark cycle[51,52]. Circadian rhythms are rhythms which occur with a periodicity of 24 h[51]. These rhythms include behavioural and physiological processes such as sleep, core body temperature, melatonin, cognitive performance and alertness[53–55]. While an intrinsically generated rhythm is maintained in the absence of external signals, environmental time cues, or zeitgebers, ensure alignment with the light-dark cycle[56]. Specifically, light cues, or photic zeitgebers, received at the eyes, are transmitted directly and indirectly to the circadian system[56]. Non-photic zeitgebers, such as ambient temperature, physical activity and social activities, also operate as time cues for the circadian rhythms, although they are a weaker signal in humans and most vertebrates[57,58].

The phase of the circadian clock can be determined by measuring physiological biomarkers such as the circulating melatonin level and body temperature, which are controlled by the circadian system and influenced by external signals such as light exposure[59–63]. Shifting sleep opportunity, even partially, into the day (e.g., while on night shifts) requires a delay of circadian rhythms[64]. Light exposure during the early morning hours of a night shift will naturally advance circadian rhythms[64], thereby working against the desired delay. Thus, managing light exposure has been the target of circadian adaptation attempts within the shift work adaptation literature[65–67].

## Mechanisms of shift work adaptation

The shift work adaptation literature has typically emphasised strategies to elicit phase delays in circadian rhythms and uses markers such as melatonin onset to measure phase shifts[59–63]. Additional markers of adaptation to shift work such as changes in health, cognition, mood and interpersonal relationships may also be worth investigating. Improvements in physiological, psychological and psychosocial outcomes, indicative of adaptation, may occur via three mechanisms of action (i.e., psycho-physiological arousal, phase shift of circadian rhythms, and efficacious coping strategies) on three different time scales across shift working careers (i.e., Within Shift, Within Block, Within Work-span) (See Fig. 1). For instance, Within Shift adaptation may result in improved alertness and performance while on shift via acute psycho-physiological arousal. Within Block adaptation may involve circadian phase shifting over consecutive night shifts (e.g., two-week blocks) and result in improvements in other markers, like sleep quality and performance. Finally, coping strategies developed by shift workers over a career (or Work-span) may assist with adaptation to the lifestyle of shift work, resulting in improved markers such as health, job satisfaction or work absenteeism. Importantly, the degree to which workers adapt to shift work is highly individualised and must be considered alongside the timeframes in which adaptation can occur[48,68].

Current approaches to shift work adaptation aiming to shift circadian phase can include pharmacological treatments[69] and the use of light blocking glasses[70] or light administering products[71]. However, evidence suggests the effectiveness of these treatments is highly variable[72]. While light is the strongest zeitgeber, numerous factors influence the efficacy of controlled lighting interventions, including patterns of exposure, such as timing, duration, frequency and intensity[72]. Logistically, it may not be possible to avoid light during the shift or the drive home and therefore optimising light

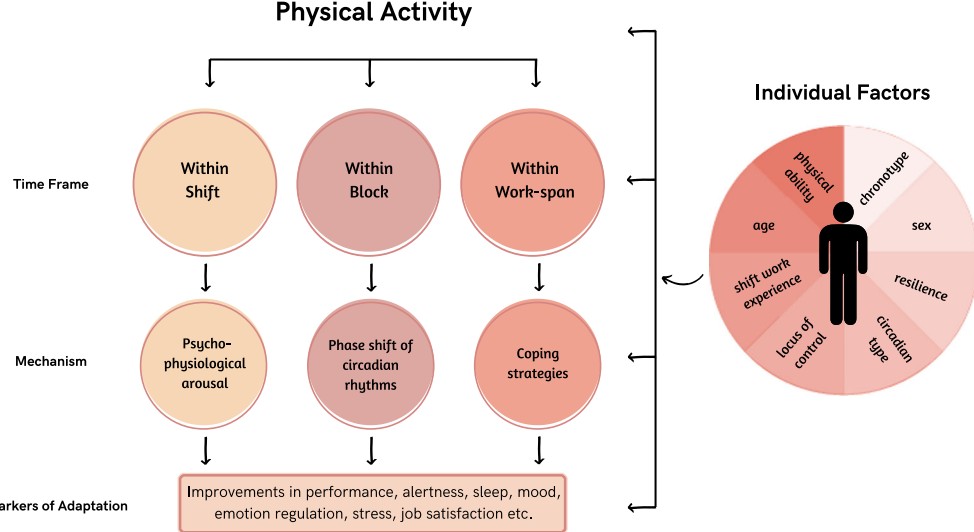

**Fig. 1 | The impact of physical activity across different shift work timeframes, and the mechanisms by which physical activity impacts adaptation.** Schematic diagram representing the impact of physical activity across different shift work timeframes, the three mechanisms by which physical activity may support adaptation (acute psycho-physiological arousal, phase shifting and long-term coping strategies), and the role of individual factors. Vector images are fair use from Canva and are not standalone content.

exposure is challenging. Further, individual, domestic and social factors can impact compliance and the effectiveness of the interventions[72]. These approaches often do not take account of the biopsychosocial influences of shift work, impeding a more holistic understanding of how to best support shift workers and industries.

A potential strategy to support all markers of adaptation across all three timeframes could be physical activity. Physical activity is a cost-effective, easily accessible behaviour that directly impacts a broad range of domains relating to cognition, physical health, and mental well-being[73–75]. Physical activity, appropriately employed, may facilitate adaptation across all three timeframes by eliciting acute psycho-physiological arousal, acting as a time cue for the circadian system, and as an enduring behavioural approach to mitigate negative consequences of long-term shift work. Thus, this paper puts forth physical activity, both during and outside of work shifts, as a potential strategy that might integrate the biological, psychological, and social aspects which help shift workers adapt to misaligned work hours (See Fig. 1).

## Within Shift

Within Shift adaptation may be experienced as immediate improvements in cognitive performance, alertness and emotion regulation during a single work shift. Physical activity may support Within Shift adaptation via acute psycho-physiological responses. Acute physical activity elicits dramatic neurophysiological changes, such as increases in neurotransmitters like dopamine and serotonin[76], and improvements in constructs such as mood and stress following acute physical activity are well documented[73,74,76–78]. Physical activity is an effective technique for self-regulation, alleviating negative mood states and enhancing positive mood states[78,79]. For instance, a meta-analysis of 158 studies examining positive mood and acute aerobic exercise found improvements in mood for low, moderate and high-intensity physical activity[78]. Larger effects were found for low-intensity physical activity, than for moderate and high-intensity physical activity[78]. These findings indicate that physical activity at all intensity levels exerts a degree of influence over positive mood states. However, it is important to note that task intensity, particularly lower-intensity activity, plays a critical role in the degree of mood improvement, a finding which is consistently echoed in the physical activity literature with other outcomes of interest, such as cognitive performance[78].

Extensive research highlights a positive effect of physical activity on post-task cognitive performance, though activity intensity may impact the endurance of post-task improvement[76,80–83]. For instance, one study explored performance of different cognitive functions in four post-activity time periods (30 min, 60 min, 90 min, 120 min) following vigorous-intensity physical activity[81]. Authors found that performance was improved for prefrontal cortex dependent tasks, like the Stroop task, but not for hippocampal dependent tasks, such as visual retention tests[81]. No performance differences were seen across the four delay groups and thus the intensity of the task potentially moderated long-term improvements in prefrontal cortex functioning. While high-intensity activity may result in enduring post-task cognitive improvements, evidence suggests that light- to moderate-intensity physical activity may result in greater overall cognitive improvement post-task[80]. A meta-analysis of acute physical activity and cognitive performance showed that tasks performed immediately following very light-, light-, and moderate-intensity exercise resulted in greater performance than high-intensity activity[80]. This effect was no longer observed when cognitive tasks were performed following a delay of more than one minute[80]. Similarly, González-Fernández et al.[84] found improvements in reaction time after exercise, though improvements were moderated by task intensity[84]. Light- to moderate-intensity physical activity resulted in greater and more consistent improvement in vigilance during the 45 min task, while vigorous-intensity saw the worst performance[84]. However, cognition was not measured post-task so the effects of task intensity on attention following a time delay in this study are unknown. Physiological responses to exercise, such as increased heart rate, endorphins, cerebral blood flow and Brain-Derived Neurotrophic Factor stimulation (a protein which is related to neuroplasticity) are all mechanisms by which cognitive performance might be improved following activity[76,80–83]. However, light-intensity physical activity immediately prior to a task may be all that is required to generate these physiological responses and elicit positive mood and cognitive performance improvements, while higher-intensity activity might be needed to see enduring benefits.

### Using physical activity Within Shift

Alongside the compounding effects of sleep restriction and circadian misalignment, physical activity intensity and shift characteristics may offer an explanation as to why physically active occupations, such as nursing, are not exempt from the cognitive declines associated with night shift. According to a systematic review of occupational physical activity in nurses, the average amount of on-shift walking per week is significantly greater than the recommended guidelines for weekly amounts of moderate-intensity

https://doi.org/10.1038/s42003-024-05962-8 **Perspective**

physical activity[85]. Day shifts were found to be more physically demanding, with day nurses spending 18% of time in moderate-intensity physical activity, and less recovery time between tasks relative to night shift nurses who spent 9% of time in moderate-intensity physical activity on shift[85]. These results might suggest that night nurses are engaging in fewer bursts of activity with less time spent in moderate-intensity activity while also contending with many other factors that play a role in the level of impairment during the night shift, like sleep restriction and circadian misalignment[85]. Thus, the lower level of physical activity in night staff may impact cognition and alertness overnight, and moderate-intensity physical activity might be required to see enduring post-activity improvements within a shift.

Consistent light-intensity physical activity throughout the night shift could promote sufficient psycho-physiological arousal to result in cognitive, alertness, and mood improvements while on shift[86,87]. A narrative review of eight studies exploring the impact of breaking up sitting with light-intensity physical activity (e.g., 3 min of walking every 30 minutes) in sedentary day workers (N = 256) found that breaking up sitting resulted in cognitive improvements in domains such as reaction time and attention[87]. It is unknown whether the effect of multiple bouts of physical activity on cognition, as a marker of adaptation, could extend to night workers experiencing circadian misalignment. Future research should explore the effect of consistent bouts of physical activity within-shift, such as a 3 min walk every 30 min, in night shift workers. This activity may be a beneficial countermeasure to support adaptation to the night shift, possibly improving employee well-being and reducing the occupational risks associated with night shift.

## Within Block

Within Block shift work adaptation refers to changes in various markers of adaptation within a block of consecutive night shifts. Consecutive night shifts, worked in mining and offshore oil installations as an example, can require up to two-week blocks of night work[40,42]. In these instances, phase shifts of circadian rhythms may indicate shift work adaptation, and be associated with improved sleep during the day, and improved cognitive performance overnight[40,88]. Strategically timed physical activity may be a non-photic time cue which entrains the circadian system within a block of night shifts, assisting adaptation through repeated exposure. Improvements in other markers of shift work adaptation, such as emotion regulation and stress, would naturally occur in line with this circadian adaptation as sleep quality and duration increases[89].

Minimum lighting requirements exist to ensure occupational health and safety in many operations[90]. Light exposure and avoidance are therefore not always feasible to assist adaptation to the night shift[72,91]. Non-photic time cues, such as physical activity, could promote Within Block adaptation to night work in these instances[57,92,93]. Evidence exists which demonstrates the effects of physical activity on circulating hormonal biomarkers (e.g., relative melatonin levels). For instance, one study of blind individuals (N = 15) without ocular photoreception, who exhibited no melatonin suppression in response to bright light exposure, found that nine individuals synchronised to the 24 h day through physical activity[93]. The authors concluded that despite non-photic zeitgebers having a weaker synchronisation effect relative to light stimuli, entrainment is possible. Similarly, physical activity at certain times elicited phase shifting effects in circadian biomarkers when the intensity and duration of activity was optimised for clock time[92,94]. The effects of one hour of high-intensity exercise at morning, afternoon, evening and nocturnal timings, on plasma melatonin secretion was examined in participants (N = 38) held under constant routine conditions (e.g., caloric intake, dim light exposure, recumbence, constant wake)[92]. The early evening exercise prior to typical dim light melatonin onset (~18:30 h) resulted in phase advances 2 h after the task though changes were no longer observed after 24 h, suggesting that early evening exercise only elicits partial adaptation of the circadian system. This partial adaptation was attributed to the continued dim light conditions which may have attenuated any observed phase advances. However, nocturnal exercise (~00:30 h) resulted in a complete phase delay in the onset of nocturnal melatonin the next day[92].

These findings contrast those of Youngstedt and colleagues[94], who saw significant phase delays in response to 1 h of moderate-intensity early evening exercise (19:00 h to 22:00 h). The authors attributed the contradictory findings to a significantly larger sample size (N = 101) that was inclusive of age and gender[94]. Taken together, these entrainment studies highlight the efficacy of physical activity as a non-photic zeitgeber, but it is important to note that lighting conditions, the time of exposure to physical activity, and the activity intensity and duration are all integral to the strength and direction of circadian phase shifting.

Emerging evidence in predominantly rodent models highlights the effects of physical activity on the timing of the molecular circadian clock in peripheral tissues[95,96]. As evidenced above, it is established that physical activity can influence behavioural rhythms and entrainment to new light-dark cycles by shifting hormonal biomarkers. However, evidence suggests that molecular circadian clocks in peripheral tissues can also respond to the timing of physical activity[95,97,98]. In rodent models, phase advances were observed in rhythmic circadian gene expression in skeletal muscle following 4 weeks of scheduled physical activity in bright light, suggesting that non-photic zeitgebers can also impact peripheral tissues[97]. However, sparse research in humans indicates physical activity has also been associated with altered rhythmic circadian gene expression[98,99]. For instance, rhythmic circadian gene expression following resistance training in one exercised leg versus the non-exercised contralateral leg, revealed that resistance training induced the upregulation of gene expression when compared to the control leg[98]. Though this study could not determine whether a phase advance or a delay occurred, these data suggest that physical activity may contribute important timing information for the synchronisation of circadian clocks in the body. Taken together, human studies of hormonal and cellular rhythms suggest that physical activity may align various circadian and peripheral clocks and could assist in the adaptation to shift work. Thus, while additional research is needed to elucidate the appropriate timing for physical activity to elicit a phase delay of hormonal and cellular rhythms in shift workers, physical activity may be a plausible strategy to assist Within Block adaptation.

## Using physical activity Within Block

Within Block physical activity interventions could be implemented in two ways: a) consistent Within Shift physical activity and b) physical activity prior to a shift. Strategically timed physical activity could help shift workers adapt their circadian rhythms to night shift work by eliciting a phase delay. Circadian phase delays would mean performance and alertness rhythms would shift, ensuring greater alertness overnight, reducing occupational health and safety risks. It is important to note that since light is the strongest driver of the circadian system, strategically timed physical activity might be most effective alongside lighting interventions in occupational environments where light exposure can be controlled. A proposed mechanism for the interaction between the two zeitgebers may be the pupillary dilation effects of physical activity[100] which enhance light perception[101]. Therefore, if light exposure or avoidance and physical activity could be timed appropriately to promote phase delays, this may be most beneficial to night shift workers.

In instances where light exposure cannot be controlled, physical activity may still provide important time cues for the circadian system. Laboratory research has shown that physical activity and bright light accelerated the entrainment of the circadian system to a new, advanced light–dark cycle when compared to bright light alone[102]. While light intensity was still controlled, this study may be closer to ecologically valid lighting conditions (i.e., constant light exposure during the shift) and provides important insight into the interaction between physical activity and unavoidable bright light on the circadian system. However, it is still unclear whether administering physical activity during the phase delay period under bright light conditions could assist circadian adaptation to the night shift. Thus, future research should consider the timing of both light and physical activity, such as bright light throughout the night and physical activity during the phase delay period. Finally, both individually managed lighting

https://doi.org/10.1038/s42003-024-05962-8                                                                                                                    **Perspective**

strategies (i.e., light administering glasses) and physical activity administered in combination during the phase delay period prior to the night shift may also be worth exploring.

## Within Work-Span

Experienced shift workers will naturally develop coping strategies and resilience across the span of a career, or Work-span[103,104]. These strategies assist in adaptation to the lifestyle of regular shift work[105,106]. Experienced shift workers are likely those who have self-selected into shift work and can adapt to work demands[107]. In contrast, those who are unable to adapt to the shift work lifestyle will likely transition into non-shift working careers where possible[108]. A key component of shift work career persistence therefore involves the use of coping strategies which help adapt workers to the working demands. In turn, these strategies support various markers of adaptation including improved work-life balance, well-being, and job satisfaction[35,105,106]. For instance, inexperienced nursing staff having worked less than one year, reported feeling increasingly socially isolated and disconnected following the commencement of shift work. In contrast, more experienced nurses reported feeling more positively supported by family and friends, alongside stronger beliefs that shift work assists work/life balance[105]. Possibly, these results may be due to senior staff having been given their preferred shift scheduling, as is the perception of many junior members[103]. Despite shift work experience and preferential bias likely contributing to differences in mood, senior nurses in this study were also more likely to have families and thus assume caring and domestic responsibilities[105]. Familial structures have been found to moderate fatigue and work strain for shift workers, and may be an important protective factor in reducing the negative vocational impact of shift work[109]. Possibly, experienced staff are better able to utilise support networks and coping strategies to facilitate long-term lifestyle adaptation across the work-span.

Chronic shift work, and night shift in particular, are considered risk factors for the onset of various health problems such as cardiovascular disease[17,75]. The mechanisms which drive this increased risk are still unknown. A potential cause for these health consequences is put forth by the Working Time Society[9] as increased exposure to 'lifestyle illnesses' resulting from unhealthy behaviours that are unrelated to circadian influences. Thus, maladaptive behaviours such as increased smoking or alcohol intake, and decreased physical activity levels, put shift workers at greater risk of disease pathogenesis[110,111]. Sleep[17] and meal timings[112] that occur out of alignment with the environmental light-dark cycle are also associated with impaired cardiometabolic function that predisposes shift workers to these conditions[9]. Strategies to align these behaviours with the light-dark cycle have benefits for cardiometabolic health. For instance, time-restricted eating is a method which shortens the eating window to optimise the timing of food intake for reduced cardiovascular disruption, and therefore reduced risk of cardiovascular disease[7,112,113]. Recent research has shown cardiometabolic benefits of a 10 h eating window in firefighters working a 24 h shift[113]. Poor sleep has also been suggested to mediate the relationship between time restricted eating and cardiovascular health[7] and as such, strategies must also consider how to improve the quality of sleep and the timing of eating for optimised cardiovascular health outcomes for shift workers. Physical activity may be complimentary to such strategies, given that the resulting acute sleep improvements[114,115] and circadian adaptation of the sleep-wake cycle[66,116] will assist the overall efficacy of other behavioural shift work strategies.

## Using physical activity within the Work-Span

Physical activity may be one countermeasure to reduce 'lifestyle illnesses' and psychological symptoms in shift workers across the work-span. A body of literature exists wherein consistently breaking up prolonged sitting with short durations of light physical activity has been shown to attenuate cardiometabolic risk markers, such as glucose and insulin responses, in physically inactive individuals[75,117–119]. In sedentary shift workers who are at increased risk of 'lifestyle illnesses', breaking up sitting throughout day and night shift may be a method to reduce the likelihood of disease pathogenesis,

and increase long-term coping with the shift work lifestyle. However, much of the breaking up sitting literature examines intervention outcomes in day working samples[75,87,117–119]. Thus, the impact of circadian misalignment on cardiometabolic health outcomes in response to a breaking up sitting intervention in shift working populations is currently unknown, though research on this is underway[120].

Shift workers may be able to use physical activity as a long-term coping strategy for the stressors of shift work, thus assisting adaptation across the work-span. The impact of activity on physical and mental health is well-documented[121]. Research highlights that consistent physical activity has important benefits for psychological outcomes such as resilience, well-being and mood[76,121–125]. Anticipating mood improvements can increase motivation for task engagement and re-engagement, and the experience of improved mood post-task can serve to strengthen habit formation[126]. Behavioural change interventions which seek to strengthen habitual tendencies through cue-behaviour associations are more sustainable for long-term behavioural change[126]. Therefore, to increase the likelihood of shift workers engaging and re-engaging with physical activity, interventions should involve habit formation principles, such as positive mood associations, to ensure increased physical activity is an enduring behavioural change. However, shift workers report varied feelings towards physical activity, with some staff prioritising recovery sleeps and socialising over physical activity following night shifts[104]. In contrast, others enjoy the flexibility to choose their own activity times[104]. Interestingly, nurses believed that physical activity during off-days improved cardiovascular fitness and enabled them to cope with night work[104]. One respondent described that they felt "more alert" on nights when they "have been out walking" and that "it's easier to stay awake…one doesn't get tired to the same extent"[104]. These differing perspectives highlight that a one-size fits all model for physical activity interventions in shift workers must consider individual perspectives on feasibility and shift worker specific cue-behaviour associations. An individualised approach within shift working contexts is vital to increase engagement with physical activity to assist adaptation to the shift working lifestyle.

Physical activity may also reduce the development and progression of psychological and physical disorders in chronic shift workers[43]. Common psychological and physical disorders such as depression, and cardiovascular, immunological and metabolic diseases in shift workers could also be related to chronic activation of central stress responses, and could be considered 'distress-related' conditions[127]. Acute stress activates both the hypothalamic-pituitary-adrenal (HPA) axis and the autonomic nervous system. The HPA axis is a central function that regulates homoeostatic processes and stress-reactive physiological responses. Increased levels of stress related hormones like glucocorticoids, such as cortisol, are activated in the HPA axis, while increases in heart rate and blood pressure occur due to autonomic nervous system activation[127]. Further, a relationship between the HPA axis and circadian rhythms exist, whereby the expression of some circadian clock genes is regulated by glucocorticoids[128]. Extrapolating from these findings, misalignment of the circadian system will therefore impact the function of the HPA axis[128]. Long-term stress activation, due to acute work stressors or circadian misalignment, without appropriate recovery time, are hypothesised causes for 'distress-related' conditions in shift workers[127].

The benefits of physical activity on health outcomes is, in part, suggested to be related to the adaptation of stress response systems. The cross-stressor adaptation hypothesis suggests that consistent physical activity supports adaptation of physiological responses to psychological stressors[129]. Physical activity shares a paradoxical relationship with stress as central stress activation is required to promote homoeostasis, though this relationship is the suggested mechanism for attenuated responses to psychological stressors[129]. Attenuated stress-reactivity might be one of the many important processes by which physical activity improves health outcomes[130]. Mixed results surround this hypothesis, with some finding reduced salivary cortisol responses in active individuals when compared to inactive individuals[130–132], equivalent decreases in cortisol activation across intervention and control groups[133], or no effect of regular exercise on cortisol response[134,135]. More

---

## Box 1 | Future research recommendations

Laboratory research should explore:
- The impact of cumulative bursts of short durations of physical activity, at all physical activity intensities, on acute changes in cognitive performance and alertness Within Shift.
- Strategically timed physical activity (i.e., during the early night prior to the night shift) at various durations and intensities on non-photic circadian adaptation within a block of night shifts. Additional outcome measures could involve non-circadian markers of adaptation, like mood, job satisfaction or work-family conflict.
- Combined lighting and physical activity approaches which involve physical activity pre-shift, and light avoidance post-shift on markers of adaptation (e.g., sleep outcomes, cognitive performance, mood states).

Research should explore:
- Employee perspectives on Within Shift, Within Block and Within Work-span physical activity, including the feasibility and perceived effectiveness.
- The relationship between naturalistic lighting (i.e., during and post-shift) and physical activity on Within Shift and Within Block adaptation of cognitive performance and circadian rhythms.

---

research is needed to further elucidate the possible stress-buffering effect of physical activity in a shift work context. If achievable, this countermeasure could have significant implications for both 'distress-related' conditions or 'lifestyle illnesses', and the development of psychological coping abilities in those with chronic exposure to shift work. Consistent exposure to physical activity might result in the adaptation of stress responses which, in turn, could facilitate adaptation across the work-span.

### Individual differences in shift work adaptation

The degree to which individuals adapt to shift work, regardless of the method used to promote adaptation (e.g., pharmacological treatments or lighting interventions), will vary due to interindividual differences. Studies demonstrate that high interindividual variability exists in the timing of circadian rhythms[136] and the sensitivity of ocular photoreceptors[101], two variables that influence the adaptation of the circadian system. Further, the level of cognitive impairment in response to sleep loss is person-specific and can be considered a trait-like characteristic[68]. One study of nurses working the night shift ($N = 162$) found that seventy-five percent of the sample received 4.7 h of sleep or less in 24 h[137] while a meta-analysis found that subjective sleep duration after night work averaged around 5 h and 51 min[138]. Thus, night shift workers are often sleep restricted during their rotations, and given the person-specific nature of cognitive impairment in response to sleep loss, performance will also likely differ between workers. In addition to intrinsic period length and light sensitivity, individual characteristics, such as circadian type and chronotype, are factors that may influence adaptation. A systematic review of 60 publications examining the relationship between individual differences and shift work tolerance, an aspect of adaptation, found that circadian type, chronotype, and locus of control are factors that influence shift work tolerance[48]. Chronotype classifications typically categorise individuals into morning, intermediate and evening types based on routine sleep schedules and preferred diurnal activities, and most people fall within the intermediate range[139]. In contrast, the revised 11-item Circadian Type Inventory (CTI) scores individuals on a languid/vigorous dimension to assess rhythm amplitude, denoting one's ability to overcome drowsiness and fatigue, particularly following reduced sleep[140]. Individuals are also categorised via flexibility/rigidity of sleeping behaviours which assesses rhythm stability[140]. Shift workers are required to profoundly alter their sleeping patterns, leaving them susceptible to drowsiness and fatigue while on shift. Chronotype is consistently studied within laboratory and field settings[141–144] with field studies demonstrating an interaction between chronotype and shift work scheduling[144]. For instance, early chronotype was associated with greater type 2 diabetes risk as shift work exposure increased, while late chronotypes experienced highest risk when working mismatched daytime schedules[144]. However, no studies have examined chronotype or circadian type within the context of

adaptation timeframes and different markers of shift work adaptation. Moreover, even less is known about how these individual characteristics interact with non-photic adaptation promoting strategies such as physical activity.

### The next steps for research and recommendations

We propose a new approach to understanding shift work adaptation, which captures additional markers of adaptation and considers the various timeframes in which adaptation may occur. Further, this approach suggests that a more comprehensive and holistic understanding of shift work adaptation can be gained from examining all markers of adaptation within shift, within block, and within work-span. This understanding may be crucial to developing occupational safety and well-being recommendations for night workers.

Physical activity might be an accessible and cost-effective health promotion strategy that could support multiple markers of adaptation across all timeframes. However, the effects of physical activity on shift work adaptation remain under-researched. Therefore, future research should explore whether physical activity influences adaptation within shift, within a block of night shifts, and within the work-span (See Box 1).

Since long durations of physical activity may be impractical while on shift, it is important to determine whether repeated doses of physical activity can improve acute cognition and mood-related aspects for shift workers while on shift. Circadian adaptation to long periods of night work is generally not feasible in naturalistic environments given that light exposure cannot be managed during or outside of work[42,145]. However, non-photic entrainment by way of physical activity may be beneficial on consecutive night shifts if the timing of physical activity can be optimised to assist overall adaptation to shift work, and the shift working schedule. Strategically timed physical activity prior to the night shift or within-shifts may assist within-block adaptation. Alternatively, these strategies may be beneficial when used in combination with lighting interventions. Circadian adaptation would result in improved sleep and psychological outcomes, mitigating the occupational health and safety risks associated with alertness during the biological night. Future research should also explore the impact of long-term physical activity on coping strategies in shift workers and if this exposure could result in reduced stress-reactivity across the work-span. The potential benefits of reduced stress-reactivity could have significant implications for conditions associated with chronic shift work, such as impaired cardiovascular and endocrine functioning[127]. Further, consistent physical activity as a coping mechanism may result in emotion-regulation abilities[125] and physical health[43], signifying long-term shift work adaptation.

It is essential to consider the practical translation of this countermeasure within industries. Factors which may limit the ability to be physically active within-shift, or outside of the workplace, such as physical health, space and time constraints must be understood. Most importantly, including shift worker perspectives on physical activity Within Shift, Within

**Perspective**

Block and Within Work-span, is essential to the implementation and sustainability of any strategy. Research suggests that employee involvement in planning, decision-making and problem-solving may be integral to improving employee interest and the success of health promotion approaches[33,146]. For instance, one study found an incongruence between company offered and staff desired worksite health promotion programmes in nurses[33]. Nurses reported that current health programmes focus on identification of health risk factors, rather than programme implementation for enduring behavioural changes[33]. Nurses suggested desired programmes could involve support groups to discuss life challenges, team building activities, and a gym in the health care facility[33]. However, barriers to the engagement in such programmes included lack of amenities (e.g., showers, nearby and free gym facilities), staff shortages, fatigue and lack of interest[33]. These results highlight a lack of employee involvement in decision making processes, which may be critical to the sustainability of health promotion programmes. Further, understanding the prioritisation of physical activity relative to sleep opportunities, family time, and rest is necessary to ensure the countermeasure is feasible and enduring[22,33]. Finally, understanding the influence of individual difference in daily activity behaviours and sleep-wake cycles, such as chronotype and circadian type, is necessary to determine the effectiveness of the countermeasure.

## Data availability
Our work is purely theoretical and did not require generation or analysis of any datasets. All relevant information can be found within the below references.

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

## Author contributions

D.F.E conceived the idea for, and wrote the first draft of, this manuscript as a component of her doctorate. D.F.E, C.C.G. and S.A.F. further developed the conceptual framework. Subsequent drafts revised and edited by D.F.E., C.C.G., G.E.V., and S.A.F. All authors reviewed and approved the final manuscript.

## Competing interests

The authors declare no competing interests.
