## [Peer Review File · Communications Biology]

Reviewers' comments:

Reviewer #1 (Remarks to the Author):

The stated objective of the review article by Easton et al is to discuss the possibility that physical activity can serve as a non-photic adaptation signal in shift workers, for improvement of sleep, cognition, health and wellbeing, as well as to decrease risk of occupational incidents. Overall, this is an interesting article, which highlights information on shift work, and proposes an interesting hypothesis/concept.

The purpose of the following comments/questions are to help the reader understand supporting evidence for the proposed hypothesis/concept.

- 1) Within the review article, the terms circadian rhythms and alignment/misalignment are frequently used. However, the Reviewer was unable to find full/accurate definitions of these terms. For circadian rhythms, the authors state that 'Circadian rhythms are any process which follows a 24 h cycle...'. The Reviewer disagrees. Circadian rhythms are considered intrinsic in nature, and should occur in the absence of external cues; this should be made clear within the review article. With regards to alignment/misalignment, please provide a definition (whether this is with respect to alignment of the organism with the environment, or between cells/organs within an organism).
- 2) The Reviewer recommends highlighting why alignment/misalignment is important, in terms of health and disease. In the absence of such background information, it may be difficult for the Reader to understand why this needs to be 'fixed' in shift workers.
- 3) The article included very few direct examples of evidence for misalignment in shift workers. For example, is there evidence that immune cell clocks do not fully phase shift in night shift workers? Perhaps mention studies by Hogenesch, who has proposed using mRNA for clock genes as a marker of circadian misalignment (PMC7441562).
- 4) The Reviewer was unable to find evidence in the article that exercise affects the timing of circadian clocks in specific tissues (as opposed to circulating markers). This becomes important when considering the use of exercise to realign circadian clocks; if exercise only affects 1 or 2 clocks in the body, the possibility would exist that exercise may worsen internal circadian misalignment (especially if other zeitgebers are out of synchrony). Perhaps consider studies by Esser and Zierath.
- 5) As mentioned in the article, a large number of zeitgebers exist, some of which tend to influence circadian clocks in tissue selective manners. As such, it may be important to align zeitgebers in an appropriate manner, for optimal circadian biology. The authors focus on physical activity and light. Another, which is likely highly relevant for shift workers, is food intake. For example, recent work by Panda and colleagues has shown benefit of time restricted feeding in firefighters.
- 6) One topic that is only briefly mentioned in the article is 'days off'. Shift work is a very heterogeneous profession, with many different types of shifts. Some can be more acute than others (e.g., 4 night shifts, 3 days off). The Reviewer is curious regarding the authors thoughts on whether it is better for shift workers to maintain (as much as possible) their night shift schedule (and therefore circadian rhythms) during their days off. The importance of this lays with the question as to whether it is better to realign circadian clocks back and forth as quickly as possible, or to maintain a constant alignment (i.e., no shift). Such a standpoint will influence 'when' exercise should be performed not only during the night shift, but also during the days off.
- 7) The topic of chronotype and shift work is fascinating. It is recommended that the author consider mentioning the Nurses Health Study results (PMC4542269).

Reviewer #2 (Remarks to the Author):

The perspective piece by Easton, Gupta, Vincent and Ferguson is an interesting and timely discussion on a potential role for physical activity to be used as a countermeasure for the negative consequences of shiftwork. The approach the Authors take, placing their hypothesis in the broader context of an appreciation of the holistic nature of circadian biology, was refreshing and certainly appropriate. Overall, there was a good consideration of pros/cons, as well as a detailed discussion of some of the potential hardships/barriers to adopting some of these approaches. The science that is cited is a mix of older and newer sources, and from this reviewers' perspective seem appropriate. My few comments below are considered minor writing changes to improve clarity and preciseness.

As this was a perspective/review piece, there was no primary data to be assessed.

Ln141 "Thus, this paper aims discusses..." seems like it should read "Thus, this paper aims to discuss..."

Ln148 I would suggest the Authors make a modification to note the circadian system is comprised of central clock in the suprachiasmatic nucleus of the hypothalamus (with some references), and many endogenous central and peripheral oscillators.

This would provide a more accurate illustration of the circadian system as a coordinated set of oscillators driven by a central clock.

Ln151 I suggest the Authors edit this section a bit as well, as it is a bit unclear. Circadian rhythms are rhythms that occur with a periodicity of about 24 h, not just "activity" which is the way this. The next thought could then be a bit clearer that examples of such rhythms in physiology and behavior include many processes such as activity, sleep, core body temperature, melatonin, and cognitive performance.

Ln160 I would add "not as strong, at least in humans and most vertebrates" since in insects and plants they can be. I know that this article focuses on humans, but it is important to make this clear for a more general audience.

Ln161 I don't really understand this sentence. Circadian timing can't be "determined" by measuring biomarkers. Maybe the Authors mean that the phase of the circadian clock can be determined by measuring biomarkers? If this is the case, then they should edit this to read that these physiological outputs are controlled by outputs from the circadian system and also by external signals... something like that would clear up things.

Reviewer #3 (Remarks to the Author):

Thank you for the opportunity to review this interesting and comprehensive review paper.

1. The paper offers a strong foundation for the need for further work in shiftwork adaptation support. More significantly, the authors propose some interesting and novel ideas about both markers and measurement frames for adaptation. Specifically, the authors propose that shift work adaptation be viewed through a holistic lens that expands to all biopsychosocial markers of adaptation. In addition, the authors propose a novel third timeframe of shift work adaptation, "within shift", setting out three timeframes in which shift work adaptation may occur and offering time-specific mechanisms and markers of adaptation. Figure 1 was particularly creative.

2. Overall, I found the paper to be well written and well supported. The arguments were well developed and logically organized. The paper was creative and offered important and novel insights and help to map out future areas of research.

I offer a few relatively minor edits and comments for your consideration below.

3. P. 4. The opening line of the paper claims "The current demands...", which is true, but it is supported by a 2003 reference which undermines the strength of your claim.

4. P. 5. Second last line. I believe you are missing a 'be'. "...some nurses consider this stress to be a strong..."

5. P. 7. This sentence needs a little work to be clear. Also see #7. "Thus, this paper aims discusses several markers of shift work.."

6. P.7 Check that you use 'timeframes' consistently throughout the paper. Sometimes it is time frames, other instances, timeframes.

6.1 P. 7 I also wonder whether it might be more useful to identify the potential strategy in the last sentence.

7. P. 7. Careful not to anthropomorphize or animate inanimate objects. For example, you indicate that "...this paper aims (or discusses?)" and later that "this paper proposes..". Instead, you might say something like "...in this paper we aim..." or "In this paper we propose..." or just 'we propose'

8. P. 8. Second sentence – check subject verb agreement.

9. Nice definitions of terms – good work here.

10. P. 8. First sentence, second par. I believe it would be more accurate to state melatonin level, as it is the relative level of melatonin that is key.

11. P. 9. Add a comma following e.g., and i.e., as these are acceptable abbreviations of: for example, and specifically, respectively, which would be followed by a comma.

12. P. 9. Last par. You claim "Current approaches" yet offer a 2014 reference for evidence, which undermines your argument. Consider how you might frame this differently.

13. Nice figure!

14. P. 14. It seems odd to shift between min and minutes within the same bracket.

15. P. 14. "A review in day workers...". Can you offer the reader a bit more information about this evidence? A review of eight X studies exploring xxx in a total of XX day workers ...?

Kind regards, Diana

Dr. Diana McMillan, RN, PhD
Associate Professor and Clinical Chair

Reviewer #1 Comments	Response to Reviewer
The stated objective of the review article by Easton et al is to discuss the possibility that physical activity can serve as a non-photic adaptation signal in shift workers, for improvement of sleep, cognition, health and wellbeing, as well as to decrease risk of occupational incidents. Overall, this is an interesting article, which highlights information on shift work, and proposes an interesting hypothesis/concept. The purpose of the following comments/questions are to help the reader understand supporting evidence for the proposed hypothesis/concept.	Thank you for taking the time to review the manuscript and for your suggestions for improvement.
#1. Within the review article, the terms circadian rhythms and alignment/misalignment are frequently used. However, the Reviewer was unable to find full/accurate definitions of these terms. For circadian rhythms, the authors state that 'Circadian rhythms are any process which follows a 24 h cycle...'. With regards to alignment/misalignment, please provide a definition (whether this is with respect to alignment of the organism with the environment, or between cells/organs within an organism).	We agree these definitions are needed throughout the manuscript. We have revised this section of the introduction to further emphasise the significant impact of shift work. Please see our response to Reviewer #2, comment #2 and #3 which have been amended to highlight the definition of circadian rhythms: Lines 165-169: "The circadian system is comprised of a central clock located in the suprachiasmatic nucleus of the hypothalamus, connected to other endogenous peripheral oscillators (Aschoff, 1960; Weaver, 1998). The cellular and molecular interactions of these central and peripheral oscillators collectively generate circadian rhythms and synchronise rhythms to the light-dark cycle (Aschoff, 1960; Kryger, 2000)." Lines 169-172: "Circadian rhythms are rhythms which occur with a periodicity of 24 h (Aschoff, 1960). These rhythms include behavioural

	and physiological processes such as sleep, core body temperature, melatonin, cognitive performance and alertness (Cagnacci et al., 1997; Dijk et al., 1992; Muck et al., 2022).” Lines 59-64 have also been amended to define circadian misalignment: “Such negative consequences of shift work result from circadian misalignment, which broadly is a term that refers to the mismatch between the timing of sleep and wake, the light dark cycle, and other circadian rhythms (i.e., behavioural, physiological, hormonal, cellular) (Resuehr et al., 2019). As such, misalignment can be internal (i.e., rhythms not aligned with each other) and/or external (i.e., rhythms not aligned with environment) which has implications for health and wellbeing.”
#2. The Reviewer recommends highlighting why alignment/misalignment is important, in terms of health and disease. In the absence of such background information, it may be difficult for the Reader to understand why this needs to be ‘fixed’ in shift workers.	We agree that this is an important addition to the paper. Please see amended lines 69-75: “The molecular and cellular impacts of circadian misalignment are thought to contribute to disease pathogenesis in shift workers (Haspel et al., 2020; Resuehr et al., 2019). The chronic inflammation of immune cells (Haspel et al., 2020), and the disruption of the regulatory role sleep plays in immune function, also impair immunity (Prather et al., 2012) and increase vulnerability to infection (Prather et al., 2015). The molecular changes arising from circadian misalignment due to shift work therefore increase risk for adverse health outcomes. ”
#3. The article included very few direct examples of evidence for misalignment in shift workers. For example, is there evidence that immune cell clocks do not fully phase shift in night shift workers? Perhaps mention studies by Hogenesch, who has proposed using mRNA for clock genes as a marker of circadian misalignment (PMC7441562).	Thank you for highlighting this and providing research to consider. This research adds an important element to the paper and has been included within the lines 65-69: “Circadian misalignment be identified through circulating circadian hormones, like the relative melatonin and cortisol levels, and also observed at the molecular level through RNA and mRNA expression (Haspel

	et al., 2020; Resuehr et al., 2019; Wu et al., 2020). For instance, one study observed significant differences in circadian rhythms of gene expression patterns between night shift and day shift nurses (Resuehr et al., 2019).”
#4. The Reviewer was unable to find evidence in the article that exercise affects the timing of circadian clocks in specific tissues (as opposed to circulating markers). This becomes important when considering the use of exercise to realign circadian clocks; if exercise only affects 1 or 2 clocks in the body, the possibility would exist that exercise may worsen internal circadian misalignment (especially of other zeitgebers are out of synchrony). Perhaps consider studies by Esser and Zierath.	The Reviewer’s comment is a very important consideration for this paper. These data add complimentary evidence for peripheral clock synchronisation. Please see amended lines 375-397: “Emerging evidence in predominantly rodent models, highlights the effects of physical activity on the timing of the molecular circadian clock in peripheral tissues (Gabriel & Zierath, 2019; Martin & Esser, 2022). As evidenced above, it is established that physical activity can influence behavioural rhythms and entrainment to new light-dark cycles by shifting hormonal biomarkers. However, evidence suggests that molecular circadian clocks in peripheral tissues can also respond to the timing of physical activity (Gabriel & Zierath, 2019; Wolff & Esser, 2012; Zambon et al., 2003). In rodent models, phase advances were observed in rhythmic circadian gene expression in skeletal muscle following 4 weeks of scheduled physical activity in bright light, suggesting that non-photic zeitgebers can also impact peripheral tissues (Wolff & Esser, 2012). However, sparse research in humans indicates physical activity has also been associated with altered rhythmic circadian gene expression (Hansen et al., 2016; Zambon et al., 2003). For instance, rhythmic circadian gene expression following resistance training in one exercised leg versus the non-exercised contralateral leg, revealed that resistance training induced the upregulation of gene expression when compared to the control leg (Zambon et al., 2003). Though this study could not determine whether a phase advance or a delay occurred, these data suggest that physical activity may contribute important

	timing information for the synchronisation of circadian clocks throughout the body. Taken together, human studies of hormonal and cellular rhythms suggest that physical activity may align various circadian and peripheral clocks and could assist in the adaptation to shift work. Thus, while additional research is needed to elucidate the appropriate timing for physical activity to elicit a phase delay of hormonal and cellular rhythms in shift workers, physical activity may be a plausible strategy to assist Within Block adaptation.”
#5. As mentioned in the article, a large number of zeitgebers exist, some of which tend to influence circadian clocks in tissue selective manners. As such, it may be important to align zeitgebers in an appropriate manner, for optimal circadian biology. The authors focus on physical activity and light. Another, which is likely highly relevant for shift workers, is food intake. For example, recent work by Panda and colleagues has shown benefit of time restricted feeding in firefighters.	The authors agree that many zeitgebers contribute to optimal circadian biology. The focus of the review is physical activity. However, we agree that highlighting the importance of optimising many other external zeitgebers is necessary to provide a comprehensive understanding of how to best support shift workers. Lines 464-476 demonstrate the adjusted section to further highlight the interaction between other aspects that support circadian adjustment: “Strategies to align these behaviours with the light-dark cycle have benefits for cardiometabolic health. For instance, time-restricted eating is a method which shortens the eating window to optimise the timing of food intake for reduced cardiovascular disruption, and therefore reduced risk of cardiovascular disease (Gupta et al., 2021; Gupta et al., 2022; Manoogian et al., 2022). Recent research has shown cardiometabolic benefits of a 10 h eating window in firefighters working a 24 h shift (Manoogian et al., 2022). Poor sleep has also been suggested to mediate the relationship between time restricted eating and cardiovascular health (Gupta et al., 2022) and as such, strategies must also consider how to improve the quality of sleep and the timing of eating for optimised cardiovascular health outcomes for shift workers. Physical

	activity may be complimentary to such strategies, given that the resulting acute sleep improvements (Kredlow et al., 2015; Wang & Boros, 2021) and circadian adaptation of the sleep-wake cycle (Boivin & Boudreau, 2014; Boudreau et al., 2013) will assist the overall efficacy of other behavioural shift work strategies.”
#6. One topic that is only briefly mentioned in the article is ‘days off’. Shift work is a very heterogenous profession, with many different types of shifts. Some can be more acute than others (e.g., 4 night shifts, 3 days off). The Reviewer is curious regarding the authors thoughts on whether it is better for shift workers to maintain (as much as possible) their night shift schedule (and therefore circadian rhythms) during their days off. The importance of this lays with the question as to whether it is better to realign circadian clocks back and forth as quickly as possible, or to maintain a constant alignment (i.e., no shift). Such a standpoint will influence ‘when’ exercise should be performed not only during the night shift, but also during the days off.	As the reviewer notes, the authors cannot provide a one-size fits all recommendation as to whether workers maintain a delayed circadian rhythm, given the many requirements outside of the work environment. In an ideal world, maintaining an adjusted rhythm for night work would be the better option for circadian adaptation, but not for other aspects of adaptation, such as improved family and social interactions. It is generally not feasible to maintain a night shift schedule because light exposure cannot be managed in or outside of work. The timing of physical activity should then be optimised to assist overall adaptation to shift work. Lines 619-624 have been amended to reflect this notion. “Circadian adaptation to long periods of night work is generally not feasible in naturalistic environments given that light exposure cannot be managed during or outside of work (Dumont et al., 2001; Ferguson et al., 2012). However, non-photoc entrainment by way of physical activity may be beneficial on consecutive night shifts if the timing of physical activity can be optimised to assist overall adaptation to shift work, and the shift working schedule.”
#7. The topic of chronotype and shift work is fascinating. It is recommended that the author consider mentioning the Nurses Health Study results (PMC4542269).	Thank you for providing further evidence to strengthen this section of the paper. We have included evidence from this study in lines 581-586 which say: “Chronotype is consistently studied within laboratory and field settings (Colelli et al., 2023; Harfmann et al., 2020; Merikanto et al., 2013; Vetter et al., 2015) with field

	studies demonstrating an interaction between chronotype and shift work scheduling (Vetter et al., 2015). For instance, early chronotype was associated with greater type 2 diabetes risk as shift work exposure increased, while late chronotypes experienced highest risk when working mismatched daytime schedules (Vetter et al., 2015).”
Reviewer #2 Comments	Response to Reviewer
The perspective piece by Easton, Gupta, Vincent and Ferguson is an interesting and timely discussion on a potential role for physical activity to be used as a countermeasure for the negative consequences of shiftwork. The approach the Authors take, placing their hypothesis in the broader context of an appreciation of the holistic nature of circadian biology, was refreshing and certainly appropriate. Overall, there was a good consideration of pros/cons, as well as a detailed discussion of some of the potential hardships/barriers to adopting some of these approaches. The science that is cited is a mix of older and newer sources, and from this reviewers’ perspective seem appropriate. My few comments below are considered minor writing changes to improve clarity and preciseness.	We thank the Reviewer for their positive comments on the manuscript and for providing constructive feedback and considerations.
#1. Ln141 “Thus, this paper aims discusses...” seems like it should read “Thus, this paper aims to discuss...”	Thank you for pointing out this grammatical issue. This has been rectified in lines 158-160: “Thus, we aim to discuss several markers of shift work adaptation...”
#2. Ln148 I would suggest the Authors make a modification to note the circadian system is comprised of central clock in the suprachiasmatic nucleus of the hypothalamus (with some references), and many endogenous central and peripheral oscillators. This would provide a more accurate illustration of the circadian system as a coordinated set of oscillators driven by a central clock.	The authors agree that this should be clarified within this paragraph to provide a comprehensive overview of the circadian system. Please see lines 165-169: “The circadian system is comprised of a central clock located in the suprachiasmatic nucleus of the hypothalamus, connected to other endogenous peripheral oscillators (Aschoff, 1960; Weaver, 1998). The cellular and molecular interactions of these central and peripheral oscillators collectively generate circadian rhythms and synchronise rhythms

	to the light-dark cycle (Aschoff, 1960; Kryger, 2000).”
#3. Ln151 I suggest the Authors edit this section a bit as well, as it is a bit unclear. Circadian rhythms are rhythms that occur with a periodicity of about 24 h, not just “activity” which is the way this. The next thought could then be a bit clearer that examples of such rhythms in physiology and behavior include many processes such as activity, sleep, core body temperature, melatonin, and cognitive performance.	Thank you for highlighting this, we have adjusted this section to ensure that this section does not misconstrue the intricacies of the circadian system. Please see adjusted lines 169-172: “Circadian rhythms are rhythms which occur with a periodicity of 24 h (Aschoff, 1960). These rhythms include behavioural and physiological processes such as sleep, core body temperature, melatonin, cognitive performance and alertness (Cagnacci et al., 1997; Dijk et al., 1992; Muck et al., 2022).”
#4. Ln160 I would add “not as strong, at least in humans and most vertebrates” since in insects and plants they can be. I know that this article focuses on humans, but it is important to make this clear for a more general audience.	We have made those adjustments to increase specificity for the broader audience. Please see line 176-179: “Non-photic zeitgebers, such as ambient temperature, physical activity and social activities, also operate as time cues for the circadian rhythms, although they are a weaker signal in humans and most vertebrates (Campbell et al., 2001; Mistlberger & Skene, 2005)”
#5. Ln161 I don’t really understand this sentence. Circadian timing can’t be “determined” by measuring biomarkers. Maybe the Authors mean that the phase of the circadian clock can be determined by measuring biomarkers? If this is the case, then they should edit this to read that these physiological outputs are controlled by outputs from the circadian system and also by external signals... something like that would clear up things.	Thank you for highlighting this lack clarity in the language. As the Reviewer proposes, this is the meaning the authors intended. The necessary changes in have been made in lines 180-183: “The phase of the circadian clock can be determined by measuring physiological biomarkers such as the circulating melatonin level and body temperature, which are controlled by the circadian system and influenced by external signals such as light exposure (Burgess et al., 2002; Eastman & Martin, 1999; Eastman et al., 1994; Smith et al., 2008; Smith et al., 2009).” And in earlier lines 130-136: “To date, these strategies have largely involved a focus on circadian adaptation. Circadian adaptation is typically defined as the process by which rhythms move to partially or fully match a new externally imposed schedule (e.g., time zone when travelling, or night shift when

	working) – otherwise referred to as phase shifting. Circadian phase shifts are defined and measured using markers of timing (e.g., onset, offset, peak, nadir) of core physiological rhythms, such as sleep/wake, melatonin, core body temperature and cortisol (Chang et al., 2013; Costa et al., 2014; Kripke et al., 2007; Roden et al., 1993)”
Reviewer #3 Comments	Response to Reviewer
Thank you for the opportunity to review this interesting and comprehensive review paper. #1. The paper offers a strong foundation for the need for further work in shiftwork adaptation support. More significantly, the authors propose some interesting and novel ideas about both markers and measurement frames for adaptation. Specifically, the authors propose that shift work adaptation be viewed through a holistic lens that expands to all biopsychosocial markers of adaptation. In addition, the authors propose a novel third timeframe of shift work adaptation, “within shift”, setting out three timeframes in which shift work adaptation may occur and offering time-specific mechanisms and markers of adaptation. Figure 1 was particularly creative.	We thank the reviewer for their positive feedback on this manuscript.
#2. Overall, I found the paper to be well written and well supported. The arguments were well developed and logically organized. The paper was creative and offered important and novel insights and help to map out future areas of research. I offer a few relatively minor edits and comments for your consideration below.	We thank the reviewer for their positive feedback on this manuscript.
#3. P. 4. The opening line of the paper claims “The current demands...”, which is true, but it is supported by a 2003 reference which undermines the strength of your claim.	We agree with the Reviewer that this sentence should be reframed and a more current reference should be included. Please see the changes in lines 51-53: “The growing demands for 24 h services mean that non-standard work hours are the norm for a significant proportion of the

	Western workforce (Folkard & Tucker, 2003; Shriane et al., 2020).”
#4. P. 5. Second last line. I believe you are missing a ‘be’. “..some nurses consider this stress to be a strong...”	Thank you for bringing drafting issues to our attention. The necessary changes have been made. See line 112: “some nurses consider this stress to be a strong factor impacting physical health”
#5. P. 7. This sentence needs a little work to be clear. Also see #7. “Thus, this paper aims discusses several markers of shift work..”	The necessary changes have been made. Please see our response to comment #1 for Reviewer #2, lines 158-160: “Thus, we aim to discuss several markers of shift work adaptation...”
#6. P.7 Check that you use ‘timeframes’ consistently throughout the paper. Sometimes it is time frames, other instances, timeframes.	We have checked throughout the manuscript to ensure consistency in language. Please see amended lines 159-160: “Thus, we aim to discuss several markers of shift work adaptation, including circadian adaptation, and three timeframes in which shift work adaptation occurs (See Figure 1)”.
#6. a. P. 7 I also wonder whether it might be more useful to identify the potential strategy in the last sentence.	We agree that this change is necessary for the flow of the review. Please see amended lines 160-162 “Further, we propose physical activity as a potential strategy which may support markers of shift work adaptation across all three timeframes (Within Shift, Within Block, and Within Work-span).”
#7. P. 7. Careful not to anthropomorphize or animate inanimate objects. For example, you indicate that “..this paper aims (or discusses?)” and later that “this paper proposes..”. Instead, you might say something like “...in this paper we aim...” or “In this paper we propose...” or just ‘we propose’	We have made the following changes to the grammar of this manuscript. Please see lines 158-162 previously mentioned. Line 146: “Thus, we propose that shift work adaptation...” and 160: “Further, we propose physical activity...” have also been amended.
#8. P. 8. Second sentence – check subject verb agreement.	The following changes have been made to improve the grammaticality in lines 169-172. Please see response to comment #3 for Reviewer #2: “Circadian rhythms are rhythms which occur over a periodicity of 24 h (Aschoff, 1960). These rhythms include many behavioural and physiological processes such as sleep, core body temperature, melatonin, cognitive performance and alertness (Cagnacci et al., 1997; Dijk et al., 1992; Muck et al., 2022)”.

#9. Nice definitions of terms – good work here.	We thank the reviewer for their kind feedback.
#10. P. 8. First sentence, second par. I believe it would be more accurate to state melatonin level, as it is the relative level of melatonin that is key.	We agree that this change will add further clarity to the paragraph. Please see these changes in comment #5 for Reviewer #2 and in lines 180-183: “The phase of the circadian clock can be determined by measuring physiological biomarkers such as the circulating melatonin level and body temperature, which are controlled by the circadian system and influenced by external signals such as light exposure”.
#11. P. 9. Add a comma following e.g., and i.e., as these are acceptable abbreviations of: for example, and specifically, respectively, which would be followed by a comma.	The appropriate grammatical changes have been made throughout the document for consistency. Please see lines 198: “(i.e., psycho-physiological arousal, phase shift of circadian rhythms, and efficacious coping strategies)” and 200: “(i.e., Within Shift, Within Block, Within Work-span) (See Figure 1) which highlight this.”
#12. P. 9. Last par. You claim “Current approaches” yet offer a 2014 reference for evidence, which undermines your argument. Consider how you might frame this differently.	We agree that this sentence does not reflect the reference year. The following changes have been made to ensure our evidence is current. Please see lines 210-212: “Current approaches to shift work adaptation aiming to shift circadian phase can include pharmacological treatments (Potter & Wood, 2020) and the use of light blocking glasses (Martin et al., 2021) or light administering products (Aarts et al., 2020).”
13. Nice figure!	We thank the reviewer for their kind feedback.
14. P. 14. It seems odd to shift between min and minutes within the same bracket.	We agree with the Reviewer. Please see those changes in line 322: “(e.g., 3 minutes of walking every 30 minutes)”
15. P. 14. “A review in day workers...”. Can you offer the reader a bit more information about this evidence? A review of X studies exploring xxx in a total of XX day workers ...?	We have made changes to lines 320-324 which provide more details from the review to improve clarity. These now read: “A narrative review of eight studies exploring the impact of breaking up sitting with light-intensity physical activity (e.g., 3 minutes of walking every 30 minutes) in sedentary day workers (N=256) found that breaking up sitting resulted in cognitive improvements in domains such as reaction time and attention”.

REVIEWERS' COMMENTS:

Reviewer #1 (Remarks to the Author):

No additional comments.

Reviewer #2 (Remarks to the Author):

The authors have done a great job addressing my comments. Thank you.

Reviewer #3 (Remarks to the Author):

Dear Authors,

I commend you for your careful and thoughtful revisions and conclude that you have made all necessary changes.

Dr. Diana McMillan